

# Evaluating quantile-based bias adjustment methods for climate change scenarios

Fabian Lehner[1], Imran Nadeem[1], and Herbert Formayer[1]

[1]Institute of Meteorology and Climatology, Department of Water, Atmosphere and Environment, University of Natural Resources and Life Sciences (BOKU), Gregor Mendel Straße 33, 1180 Vienna, Austria

**Correspondence:** Fabian Lehner (fabian.lehner@boku.ac.at)

**Abstract.** Daily meteorological data such as temperature or precipitation from climate models is needed for many climate impact studies, e.g. in hydrology or agriculture but direct model output can contain large systematic errors. Thus, statistical bias adjustment is applied to correct climate model outputs. Here we review existing statistical bias adjustment methods and their shortcomings, and present a method which we call EQA (Empirical Quantile Adjustment), a development of the methods ED-

CDFm and PresRAT. We then test it in comparison to two existing methods using real and artificially created daily temperature and precipitation data for Austria. We compare the performance of the three methods in terms of the following demands: (1): The model data should match the climatological means of the observational data in the historical period. (2): The long-term climatological trends of means (climate change signal), either defined as difference or as ratio, should not be altered during bias adjustment, and (3): Even models with too few wet days (precipitation above 0.1 mm) should be corrected accurately,

so that the wet day frequency is conserved. EQA fulfills (1) almost exactly and (2) at least for temperature. For precipitation, an additional correction included in EQA assures that the climate change signal is conserved, and for (3), we apply another additional algorithm to add precipitation days.

## 1 Introduction

Daily data from climate models are used for various applications, e.g. in hydrology, silviculture and for general climate risk

studies (e.g. Horton et al., 2017; Seidl et al., 2019). However, simulated outputs from global climate models (GCMs) and regional climate models (RCMs) can exhibit large systematic biases relative to observational data sets (Mearns et al., 2013; Sillmann et al., 2013). Such systematic errors can be statistically adjusted with gridded observations. Those adjusted data sets are widely used (e.g. Bao and Wen, 2017; Thrasher et al., 2012; Chimani et al., 2016) but are controversial due to various errors introduced by statistical adjustment. In the last two decades a series of methods for statistical bias adjustment have been

presented.

Simple methods that correct the mean and/or the variance of the model data have been introduced (Maraun, 2016; Lafon et al., 2013; Widmann et al., 2003) and are still in use due to their simplicity (Navarro-Racines et al., 2020).

Models may have different biases for extremes than for average values (Di Luca et al., 2020a, b). To improve the distribution of meteorological variables, more sophisticated approaches have been introduced. They adjust every quantile of the cumulative





distribution functions (CDF) according to the differences between daily modeled and observational data during a reference
period. There are many different variations and names for this method in the literature: variable correction method (Déqué,
2007), distribution-based scaling (Yang et al., 2010; Seaby et al., 2013), distribution mapping (Teutschbein and Seibert, 2012),
statistical bias correction (Piani et al., 2010), statistical transformation (Gudmundsson et al., 2012), quantile-quantile mapping
(Hatchett et al., 2016; Potter et al., 2020; Charles et al., 2020) or quantile mapping (QM) (Lafon et al., 2013; Themeßl et al.,
2011; Maraun, 2016).

The distribution of meteorological variables can be described with empirical CDFs which is a non-parametric approach
(e.g. Cannon et al., 2015). Many QM methods use a parametric approach instead (e.g. Hempel et al., 2013; Piani et al., 2010;
Switanek et al., 2017), where statistical functions such as gamma or normal distributions are fitted to the CDFs. Another
parametric method is the multi-segment parametric QM like MSBC (Grillakis et al., 2013, 2017). The CDFs of both model
and observations are approximated by piece-wise functions which can better represent the original CDF than one single fit to
the entire CDF. Whether to use a non-parametric or a parametric approach is still in scientific discussion (Teng et al., 2015) but
the non-parametric approach is more common. Lafon et al. (2013) compared non-parametric (empirical) and parametric QM
and found that the empirical approach was the most accurate. Cannon et al. (2015) and Gudmundsson et al. (2012) also prefer
the empirical QM. Themeßl et al. (2012) point out that parametric QM can introduce new biases, because the distribution of a
meteorological variable is not fully known and also depends on the region and season. However, non-parametric QM depends
more on the calibration period than parametric QM. Switanek et al. (2017) argue that the correction of extremes is more robust
with a parametric approach, as the return level of the most extreme event is somewhat random. This can be improved by fitting
function to the distributions. Examples for the classification of methods as parametric or non-parametric is shown in Table 1.

One feature of traditional QM is that it may alter the raw climate change signal 'CCS' found in the model (i.e. the change
of the arithmetic mean of a meteorological variable over time) (Hagemann et al., 2011; Maurer and Pierce, 2014; Maraun,
2013, 2016). This may sometimes be a desired feature as the CCS of the model itself could be biased (Boberg and Christensen,
2012; Gobiet et al., 2015). In some cases, it has been argued that CCS-changing bias adjustment methods may even improve
implausible trends (Maraun et al., 2017). Especially, if the model has large errors in circulation patterns, CCS-preserving bias
adjustment may amplify the bias (Maraun et al., 2021) and thus lead to implausible trends.

This means, that the choice of a climate model with plausible weather patterns and a plausible CCS is crucial (Maraun et al.,
2021). If the trend simulated by the model is trustworthy (see Chapter 12 in Maraun and Widmann (2018) for further discussion
on this topic) one might want to keep the trend unchanged after bias adjustment. As a workaround for trend preservation, Bürger
et al. (2013) and Hempel et al. (2013) removed the trend before QM and added the trend back again after bias adjustment
(detrended quantile mapping - DQM). A trend-preserving method termed as quantile delta mapping (QDM) was developed
by Cannon et al. (2015). It was implemented as a parametric, slightly changed method by Switanek et al. (2017) who named
their approach scaled distribution mapping (SDM). A very similar approach termed the equidistant CDF matching method
(EDCDFm) was introduced by Li et al. (2010) which was later improved by Pierce et al. (2015). Cannon et al. (2015) prove in
their appendix that EDCDFm and QDM are equivalent in the end, however different they are in concept.





Bias adjustment methods which do not alter the CCS implicitly assume time-invariance (Maraun and Widmann, 2018) for
the bias, i.e. that the mean bias is time independent and therefore the predicted trends are credible. Methods that do not alter
the CCS include EDCDFm, MBSC (Grillakis et al., 2013, 2017), QDM, PresRat (Pierce et al., 2015) and SDM. In the end, all
five of these methods assume that the biases at quantiles do not change over time (overview in Table 1).

Note that the definition of the time-invariance assumption, sometimes also called stationarity assumption (Switanek et al.,
2017) is inconsistent in the literature. For example, Switanek et al. (2017) state that standard QM has the underlying assumption
of time-invariant stationarity contradicting Maraun and Widmann (2018). We assume these inconsistencies come from different
definitions. Stationarity can refer to a time-independent mean bias, or to a time-independent bias found at certain absolute
values of a meteorological variable. Since QM does alter the CCS we conclude that QM implies that the mean bias changes
over time.

EDCDFm is always capable of preserving the CCS in the median (and also at every quantile). If applied additively this also
holds true for the arithmetic mean. For precipitation, a multiplicative approach is more suitable. Pierce et al. (2015) call the
multiplicative method PresRAT; it preserves the model predicted CCS in median (and also at every quantile) but not the mean
CCS. It may make sense to correct the mean CCS on a monthly, seasonal or annual basis after bias adjustment (Pierce et al.,
2015).

Most of the bias correcting methods correct a wet day bias of a climate model (i.e. the number of wet days above a specific
precipitation threshold) only if the model has a positive wet day bias. However, in some rare cases the model may have too
few wet days. Often, a multiplicative bias adjustment is selected for precipitation (e.g. Switanek et al., 2017; Pierce et al.,
2015; Cannon et al., 2015). To avoid division by zero during bias adjustment, dry days have to be treated separately. Only few
studies have focused on correcting a negative wet day bias, with one of them being Themeßl et al. (2012). They use a simple
linear interpolation to fill the gap of wet days in the precipitation CDF. This does not necessarily conserve precipitation sums,
because the CDF of precipitation does not follow a linear curve. Some authors solved this problem by modifying the dry days
prior to bias adjustment (Cannon et al., 2015; Cannon, 2018; Mehrotra et al., 2018; Vrac et al., 2016).

There is no single best bias adjustment methods that fits all needs. The advantages and disadvantages of the bias adjustment
methods mentioned here depend on the application. Maraun and Widmann (2018) and Doblas-Reyes et al. (2021) compre-
hensively review the whole topic, the motivation behind bias adjustment and the historic development. Generally speaking,
distribution based methods like QM usually outperform other simpler methods like mean bias adjustment as shown by Lafon
et al. (2013) or Themeßl et al. (2011). In Pierce et al. (2015) EDCDFm is preferred over QM because it does not alter the
CCS. Casanueva et al. (2020) tested SDM, DQM, QDM, an empirical and a parametric QM and others and concluded that
trend-preserving methods like SDM and DQM are preferable. Large comparative studies (Maraun et al., 2019; Gutiérrez et al.,
2019; Widmann et al., 2019) provide an overview of many bias adjustment methods.

All methods above correct each variable independently, usually on daily data and on a single grid cell and therefore belong
to the group of univariate bias adjustment algorithms. Nevertheless, all of them significantly improve the spatial patterns of
climatological data depending on the observational data. When it comes to smaller time scales (e.g. a season, a month or a
single day), many of them are still able to improve spatial patterns compared to raw model data (Widmann et al., 2019) and





the temporal variability of model data (Maraun et al., 2019) to some extent. However, methods have been developed to address
these joint aspects. These go beyond the scope of this paper but are worth mentioning:

Some authors introduce methods to correct the temporal autocorrelation across several days, weeks or months (Nguyen et al., 2016, 2017; Pierce et al., 2015; Mehrotra and Sharma, 2016). When several time scales are corrected one after the other, this is referred to as nesting approach. A different approach to improve temporal statistics was introduced by Volosciuk et al. (2017) with a two-step approach. It consists of QM on the model's spatial resolution in a first step and a downscaling with a stochastic
regression-based model as a second step which adds random small-scale variability. However, the added skill is different from case to case and may even increase the bias at times.

Other authors find the results of univariate methods for spatial precipitation patterns on specific days in the model unsatisfactory (Pastén-Zapata et al., 2020; Potter et al., 2020; Charles et al., 2020). Spatiotemporal statistics like the plausibility of weather patterns can be improved by correcting across multiple time scales and variables (even though on a single grid cell)
as shown by Mehrotra and Sharma (2016), Mehrotra et al. (2018) and Mehrotra and Sharma (2019). However, multivariate methods suffer from disadvantages such as very high computational demands or a limited measure of the full multivariate dependence of structure (e.g. Cannon, 2018; Bürger et al., 2011).

The goal of this paper is to find a suitable quantile based bias adjustment method that could be used for climate impact modeling studies which are sensitive to the changes in means to thresholds effects. We choose to focus only on quantile-based
methods, because they usually outperform simpler methods, as described above. We posit that three important demands should be met:

- (1): The bias adjusted data should match the observational data in the historical period in terms of arithmetic mean.

- (2): The CCS should not be altered during bias adjustment. In other words the mean change between historical and simulated future period from the raw model should be preserved. This should also hold true for the ratio of the CCS, if
the bias adjustment is applied multiplicatively.

- (3): Models with too few wet days should be corrected reasonably which means that a way has to be found to add wet days.

Almost all quantile-based corrections are capable of meeting demand (1) for temperature, but for precipitation new challenges arise: First, parametric methods struggle to find an accurate function for daily precipitation values and second, demands
(1) and (3) are linked, i.e. if only wet days are bias adjusted in a model with too few wet days, the adjusted precipitation sum in the historical period will be also too low. Inaccurately corrected model data can cause wrong conclusions from climate impact studies, when the future climate data is compared to the historical data. For demands (2) and (3), we present and apply additional methods (Section 3.2 and 3.3).

In this paper we compare how well certain quantile-based bias adjustment methods meet these three demands. SDM is
selected because it has been used for larger projects in Austria (Chimani et al., 2016, 2019), it outperforms other methods (Casanueva et al., 2020) and it is a parametric method. Traditional empirical QM is widely used and is part of many comparison



studies (Widmann et al., 2019; Maraun et al., 2019; Pierce et al., 2015; Casanueva et al., 2020; Smith et al., 2014). As a third method we developed a combination of PresRAT (Pierce et al., 2015) and EDCDFm (Li et al., 2010) as they show promising results with respect to the CCS. We unite both methods but apply them in an explicitly empirical (non-parametric) manner

as we experienced problems with fitting functions to the CDF of daily precipitation values (Vlček and Huth, 2009). As the empirical aspect is an important feature of this new approach, we named the method EQA (Empirical Quantile Adjustment). Table 1 shows a classification of quantile-based bias adjustment methods. The methods in bold are used in this paper.

**Table 1.** Grouping of some quantile-based bias adjustment methods in two categories. Note that this list is not complete. The methods in bold are used in this work.

|  | Parametric | Non-parametric / Empirical |
|---|---|---|
| Bias at fixed quantile / Trend preserving | EDCDFm (Li et al., 2010) <br> MBSC (Grillakis et al., 2013, 2017) <br> PresRAT (Pierce et al., 2015) <br> **SDM (Switanek et al., 2017)** | **EQA (Section 3.1)** <br> QDM (Cannon et al., 2015) |
| Bias at fixed value / Trend altering | DQM (Hempel et al., 2013) <br> QM (Piani et al., 2010; Lafon et al., 2013) | **QM (Lafon et al., 2013; Themeßl et al., 2011)** <br> QUANT (Gudmundsson et al., 2012) |

## 2   Data and area of interest

This study focuses on Austria which is located in Central Europe and is representative for a mountainous area in the middle

latitudes. The topography is shown in Fig. 1. A large part of the Eastern Alps are within the Austrian borders. The elevation ranges from 114 m in the East of Austria to 3798 m amsl on the highest mountain. Because of the complexity of the topography, the spatial resolution of GCMs and also RCMs is not sufficient to resolve mountain ridges and valleys. The climatological properties can change within a few kilometers due to topographically induced effects (Stauffer et al., 2017).

Austria has a large number of high quality weather observation stations that are operated by Zentralanstalt für Meteorologie

und Geodynamik (ZAMG). Also, gridded observational data sets called SPARTACUS for minimum temperature, maximum temperature and precipitation are available on a daily basis at a high spatial resolution of 1 km (Hiebl and Frei, 2016, 2018). The time span reaches from the year 1961 to 2019. SPARTACUS mostly uses stations with long time series to provide robust trends for climate change.

For the observational data, SPARTACUS in its unchanged form is used (hereafter named OBS). For model data, synthetic

data is produced by smoothing SPARTACUS data with a running mean of 12 km. This is a typical spatial resolution of RCMs.




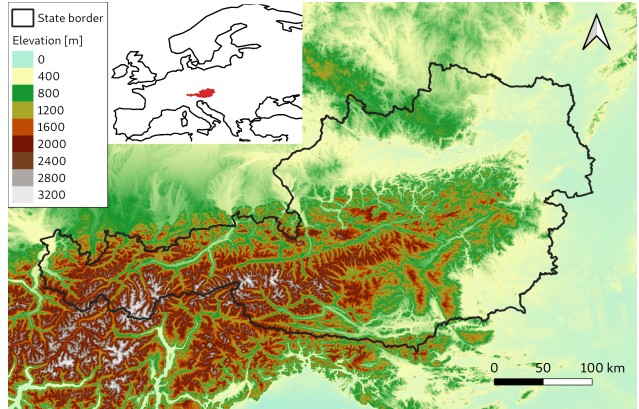

**Figure 1.** Area of interest with Austrian state borders. © European Union, Copernicus Land Monitoring Service 2020, European Environment Agency (EEA).

To generate artificial data with loo few wet days and too little precipitation, the data was further manipulated. This was done by multiplying the precipitation of each day with a uniformly distributed random number between 0 and 1. Furthermore, a trend to even drier conditions was introduced by successively canceling more and more wet days going from 1961 to 2019.

To show that the bias adjusted model data do not always match the observations in the historical period, we analyzed data

sets in Austria from the projects ÖKS15 (Chimani et al., 2016) and STARC-Impact (Chimani et al., 2019). In both projects, the model data was bias adjusted with SDM (Switanek et al., 2017). This data is freely available via the Climate Change Center Austria (CCCA) and consists of bias adjusted temperature and precipitation data from several RCMs at a spatial resolution of 1 km. The data is used for many climate impact studies in Austria (e.g. Jandl et al., 2018; Unterberger et al., 2018).

We calculated climatological annual precipitation sums for all models in ÖKS15 and STARC-Impact in the reference period

1971-2000 and for the observation data set GPARD1 for the same time period (e.g. Chimani et al., 2016; Hofstätter et al., 2015). This period was used for bias adjustment in the two projects. The bias for each model in the period 1971-2000 is calculated as the difference between the mean of models and observation. Fig. 2a shows the bias of the domain average annual precipitation for each model. The mean bias ranges from approx. -6 % for the driest model to +2 % for the wettest model. The comparison on a grid cell basis on the right side in (Fig. 2b) shows biases of more than 5 % for the wettest 0.1 percentile, and a bias of

approx. -25 % for the driest grid cells. However, the median bias of all models is +0.5 % which we consider as quite good.

Looking further into all the models used in ÖKS15 and STARC-Impact, we found that the largest errors occur in very dry models with a distinct negative wet day bias. Therefore, we focus on the bias adjustment of very dry climate models in this paper.

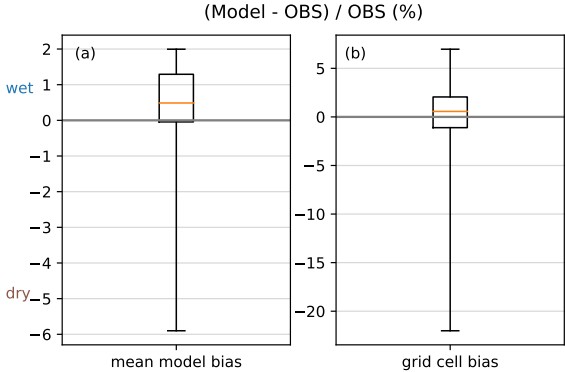

**Figure 2.** Box and whisker plot for the relative annual precipitation bias (%) of the ÖKS15 and STARC-Impact models (a total of 35 models) to the observational data set GPARD1 for the reference period 1971-2000. A positive bias indicates that the model is wetter than the observations. (a): Relative bias for the area mean for each climate models in Austria. (b): Relative bias on a grid cell basis. The upper (lower) whisker shows the 99.9 (0.1) percentile. The box ranges from the 25 to the 75 percentile, the orange horizontal line shows the median.

## 3   Methods

This study focuses on implementing EQA to bias correct data from climate models and compares it with two existing methods, namely QM and SDM. To preserve the relative CCS of precipitation, an additional algorithm equivalent to that in Pierce et al. (2015) adjusts the CCS (Section 3.2).

All three methods are quantile-based bias adjustment methods that adjust the climate model data to match the CDF of the observation. The daily data of each grid cell of the model is adjusted separately with the observations on a monthly basis. For

the calibration data, a time period of 30 years is typical, since the statistical distribution of data of a shorter time period can be very noisy and a longer time period usually has pronounced climatological trends.

### 3.1   Empirical Quantile Correction (EQA)

EQA is based on EDCDFm (Li et al., 2010) for adjusting temperature and on PresRAT (Pierce et al., 2015) for correcting precipitation but in contrast to these it is purely non-parametric, as we experienced problems with fitting functions to daily

precipitation values (Vlček and Huth, 2009).

The basic assumption is that the model bias remains constant over time for each quantile of the model data. In other words, we consider that the RCM is able to predict a ranked category of temperature or precipitation but not the value for this variable (Déqué, 2007). This especially holds true if the frequency of weather patterns does not change significantly over time where certain quantiles are often linked to certain weather patterns. A specific weather pattern will have different absolute values

in future but the same quantile. In other words, EQA implies that there are only small changes of weather patterns over time because larger changes would change the quantiles of weather patterns. This is conceptually very different from the approach of QM where the bias is expected to stay constant at fixed values which means that the bias of fixed, absolute values are time





independent (see introduction). We argue that the bias of a model is more likely to stay constant for certain weather situations than at fixed values in the context of climate change. Our approach of correcting quantiles is also supported by Maraun and

Widmann (2018) who state that biases depend not only on the actual values but more generally on the state of the climate system. The advantage of constant biases at quantiles is also shown by the following example:

Consider the daily maximum temperatures for a grid point during a summer month in Europe, where three quantiles of the observations in the reference period are 20, 25, and 30 °C. The model simulates 20, 30 and 32 °C for the same period, i.e. there is a warm bias especially in the middle quantile. QM would suggest 0, -5 and -2 °C as correction values for the model values.

For the future period the model simulates 25, 35 and 36 °C. QM would correct this 22.5, 33 and 34 °C. In this example, values in between are linearly interpolated, values above the range in the reference period (above 32 °C) are found through constant extrapolation, that is the correction value for the highest temperature also applies for even higher temperatures. EQA corrects at quantiles (not fixed values) which yields to 25, 30 and 34 °C. In this simple example, EQA seems to plausibly correct the model's warm bias at middle quantiles, while QM does not.

To make EQA easily accessible, we describe our approach step by step. The procedure is divided into two parts. In the first part (steps 1 to 3), the correction values (CVs) are evaluated for a distinct number (e.g. 100) of quantiles of a variable's CDF. In the second part (steps 4 to 5), the correction values are applied to any desired time interval.

Step (1): If the variable is not limited to non-negative values (like temperature or dew point), detrending should be applied to the data before any further calculation is done. Trends may otherwise artificially increase the variance of the data, which

should be avoided. The 30 year data is detrended for each month separately by subtracting a linear trend. This trend is added again after bias adjustment. Removing a linear trend is just a first order approximation of a general trend. Detrending could also be done with polynomials of higher order but for simplicity we assume a linear trend.

Step (2): If the variable of interest is precipitation, the further procedure depends on the difference in wet days between the model and the observation. If the model data has more wet days, all data is used. If the model data has less wet days than the

observational data (which is quite rare), then only wet days are used for further calculation to avoid division by zero in Eq. 4. The threshold for wet days is typically set at 0.1 mm precipitation per day.

Step (3): The mathematical description is similar to EDCDFm in Eq. 2 in Li et al. (2010). The starting point is the second and third term on the right side of this equation which is the term that corrects the raw model data:

$$x_{corr} = x_{m-f} + \underbrace{F_{o-c}^{-1}(F_{m-f}(x_{m-f})) - F_{m-c}^{-1}(F_{m-f}(x_{m-f}))}_{\text{correction term}}. \tag{1}$$

where $x_{corr}$ is the time series of the corrected variable, $x_{m-f}$ is the original time series of the variable, F is the CDF of either the observations (o) or model (m) for a historic (calibration) period (c) or future (projection) period (f). F is an empirical function in this and all following equations. For EQA, the terms in brackets $(F_{m-f}(x_{m-f}))$ are now replaced by $F_{100}$ which





consists of 100 equidistant values from 0.5 % to 99.5 %, which are the 100 CVs to correct the model data:

$$F_{100} = \begin{pmatrix} 0.995 \\ 0.985 \\ ... \\ 0.015 \\ 0.005 \end{pmatrix} \tag{2}$$


$$\mathrm{CV} = F_{o-c}^{-1}(F_{100}) - F_{m-c}^{-1}(F_{100}). \tag{3}$$

The number of 100 points seems to be a reasonable compromise. A higher number would be less robust to extremes, as especially the CVs of extremes would depend even more on single extreme events. A lower number would provide less detail about the distributional shape of the model bias. In Eq. 3 the CVs are defined as the difference between two inverse CDFs
which is used for variables such as temperature and dew point.

For parameters that have a meteorologically meaningful zero value as a lower boundary, a multiplicative approach is more useful, e.g. for precipitation, wind speed or global radiation (see also Pierce et al., 2015). The CVs for those parameters are found by

$$\mathrm{CV} = \frac{F_{o-c}^{-1}(F_{100})}{F_{m-c}^{-1}(F_{100})} \tag{4}$$

If the model wind speed and global radiation should ever reach exact zero in the denominator, Eq. 4 would not be defined. For this case the corresponding CVs are manually set to 0. For precipitation, one has to distinguish between models with too many or too few wet days (described in step (2) above, further information in Sect. 3.3). The CVs can be interpreted as the model bias for each quantile of the model data at a given grid cell.

Step (4): Any desired time period for bias adjustment is selected (future or historical). It is possible to choose the calibration
period itself. Again, if the variable is not limited to non-negative values, the linear trend has to be removed from the model data to avoid altering the CCS and added back after bias adjustment. The time period to be chosen is usually a 30-year period, as for the calibration time period.

Step (5): The CVs are added (Eq. 4 for temperature and dew point) or multiplied (Eq. 5 for precipitation, global radiation and wind speed) to the selected (e.g. future) model data $x_{m\text{-}f}$. This results in the bias corrected data $x_{corr}$. Mathematically,
this can be described as:

$$x_{corr} = x_{m\text{-}f} + \mathrm{CV}_i \tag{5}$$

$$x_{corr} = x_{m\text{-}f} \cdot \mathrm{CV}_i \tag{6}$$





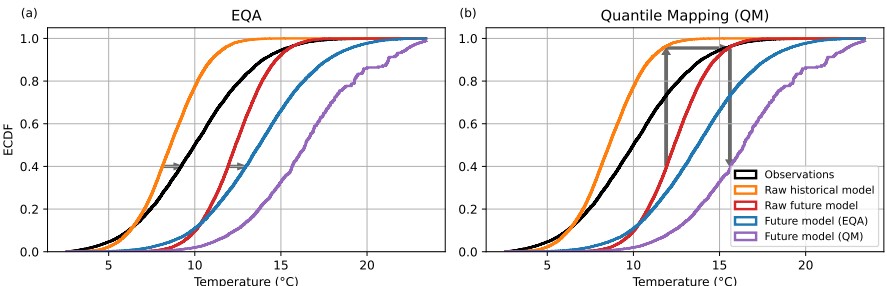

**Figure 3.** Schematic of bias adjustment for temperature data. CDFs are shown for following data: Observational data (black), raw historical model (orange), raw future model (red), future model corrected with EQA (blue) and future model corrected with QM (purple). The arrows illustrate the bias adjustment path for future model data. Panel (a) shows EQA, where the model bias in the calibration period (left arrow) is applied to the future model data (right arrow that has the same length as the left arrow). Panel (b) shows QM, where the correction value is found at the model bias in the historical period of the same absolute value.

where $x_{m\text{-}f}$ is the ranked future model data and the CVs are interpolated to $\text{CV}_i$ to match the length of $x_{m\text{-}f}$. This ensures that

every value of $x_{m\text{-}f}$ is matched with the CV of the same quantile. For values within the range of $F_{100}$, the CVs are linearly interpolated. For extreme data at both ends of the distribution, the CVs have to be extrapolated. This is done via constant extrapolation, i.e. the first (last) CV is used for correcting data below (above) the outermost CDF value. All model data values below the 0.5 % percentile are corrected with the CV attached to the 0.5 % percentile and model data values above the 99.5 % percentile are corrected with the CV attached to the 99.5 % percentile. As the result $x_{corr}$ is also ranked, the values have to be

rearranged in the original order in the time series.

Step (6): For a smooth transistion between different adjustment periods, not all 30 years that were used during bias adjustment are saved, but only the middle 10-year period. For this reason, the bias adjustment has to be calculated in 10-year steps.

The graphical solution for bias adjustment for temperature data is shown in Fig. 3. The temperature data in this plot is

artificially created following a normal distribution, where the historical period is 1981-2010 and the future period is 2071-2100. In this hypothetical case the present-day model has a significant bias in mean and variance:

– The observational data features a mean of 10 °C and a standard deviation of 3.1 °C.

– The raw historical model (1981-2010) has a cold bias in the data with a mean of 8 °C and a standard deviation of 1.8 °C.

– The raw future model (2071-2100) is warmer with a mean of 12.4 °C but the standard deviation remains unchanged to

the historical model with a standard deviation of 1.8 °C.

For the raw historical model, these distributions result in too low temperatures at the upper end of the CDF and slightly too high temperature at the lower end. During the bias adjustment of EQA, this model bias of each quantile is added to the future model resulting in the bias corrected model (purple line). In contrast, QM uses absolute model values from the historical





period. As during climate change higher temperatures occur more often, correction values from the upper part of the CDF are
used more often. As higher temperatures tend to have larger biases in the raw historical model, the adjustment with QM results
in a bias adjusted model that is too warm.

### 3.2 Precipitation: conserving the CCS

EQA (Section 3.1) conserves the raw model's CCS on each quantile. For additive EQA (used for temperature and dew point),
this is also valid for means and sums. However, multiplicative EQA does not conserve the relative CCS of means for precipita-
tion. As the precipitation sum (monthly sum, annual sum) is usually more important than the precipitation at a specific quantile
in the CDF, an additional algorithm is developed to reproduce the raw model's change in means. This is referred to as the
conservation of the model CCS. Depending on the application of the corrected precipitation data, one can adjust the monthly,
seasonal or the annual CCS. The following approach is equivalent to the ones in Pierce et al. (2015) and Charles et al. (2020).

For a future time period, the CCS for precipitation for the raw model for one grid point is

$$\text{CCS}_m = \frac{\overline{R_{m\text{-}f}}}{\overline{R_{m\text{-}c}}} \tag{7}$$

where $\overline{R_{m\text{-}f}}$ is the mean precipitation of the model in the future time period and $\overline{R_{m\text{-}c}}$ is the mean precipitation of the model
in the historical (calibration) time period. The mean is either a monthly or annual climatological mean. The CCS for the bias
adjusted data after EQA is

$$\text{CCS}_{corr} = \frac{\overline{R_{corr\text{-}f}}}{\overline{R_{corr\text{-}c}}} \tag{8}$$

The error $E$ of the CCS of the corrected model compared to the CCS of the raw model (in %) is defined as

$$E = \frac{\text{CCS}_{corr}}{\text{CCS}_m} \cdot 100 - 100, \tag{9}$$

where a value of 0 is a perfect bias adjustment method. The precipitation (daily data) of the bias adjusted model data $R_{corr\text{-}f,\,t}$
for every day t is corrected with

$$R_{corr\,CCS,\,t} = R_{corr\text{-}f,\,t} \cdot \frac{\text{CCS}_m}{\text{CCS}_{corr}} \tag{10}$$

to match the CCS of the raw model data. Equations 7 and 8 can be applied for either monthly, seasonal or annual data or for
both, applied one after the other. However, every CCS cannot be exactly conserved at the same time, because the second CCS
(e.g. the annual one) alters the data from the first CCS correction (e.g. monthly).

### 3.3 Precipitation: adding wet days after EQA with a piecewise trapezoid approach (EQAd)

EQA (along with other methods like QM and SDM) corrects by default the number of wet days if the model has more wet
days than the observational data by multiplying the lower parts of the model CDF by 0. However, EQA cannot add wet days
that are initially not in the model. Therefore, an extension of EQA called EQAd (d for dry mode) is presented that adjusts the





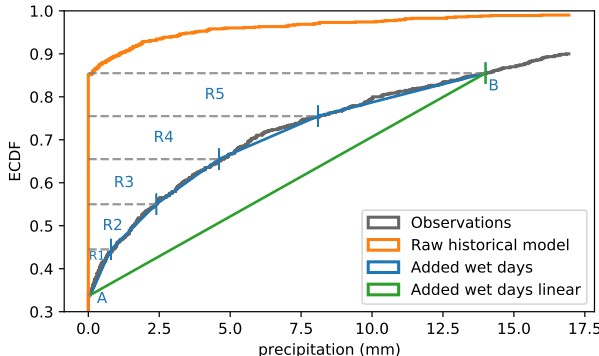

**Figure 4.** Schematic of two different methods to add wet days in the ECDF of precipitation. Themeßl et al. (2012) presented a simple linear method (green line) that leads to too high precipitation sums and is only suitable if the wet day bias of the model is low (below 10 % of total days). A more sophisticated method (blue lines) leads to very accurate precipitation sums that match the observations. The horizontal dashed gray lines are equidistant in y-direction and indicate the trapezoids, that are used for the calculation of piece-wise precipitation sums.

wet day frequency of the model to match the observational wet day frequency via a piecewise trapezoid approach. Also, the amount which is added should reproduce the precipitation sum of the observational data.

The model bias in wet day frequency is defined as an absolute one, i.e. the number of wet days that have to be added to the
model data. This is more robust for extremely dry models or extremely dry climates.

For wet days that have to be added, the simplest approach is to draw a straight line (green line in Fig. 4) between the lower point in the CDF, where wet days start in the observations (A in Fig. 4) to the point where the wet days start in the model at the corresponding precipitation value of the observations (B in Fig. 4). This approach was chosen by Themeßl et al. (2012). However, the shape of precipitation in an ECDF is far from linear. Thus, this method only reproduces the wet day frequency of
the observations but not the precipitation sums. In fact, it produces more precipitation than the observed precipitation. However, this method can deliver satisfactory results if the model bias in wet days is relatively low. We set the threshold to 0.1 (i.e 10 % of all days) in the ECDF for this linear approach.

To preserve the precipitation sum of the observations for a bias of wet days of more than 10 % of all days, wet days can be added by piece-wise linear interpolation with the constraint that the precipitation sum has to match the observations in
the historical period. As this method ought to be applicable in future as well, the missing precipitation sum is defined as a relative quantity, i.e. the ratio of the missing precipitation in the lower part of the CDF to the whole precipitation (observational precipitation above 0.85 in the CDF in Fig. 4). This relative quantity can also be used for future time periods, as the precipitation sum of the upper part of the CDF can be calculated from the model after bias adjustment with EQA. The algorithm to add wet days should be applied between steps (2) and (3) during the EQA method in Sect. 3.1.
The presented trapezoid method adds wet days after bias adjustment. Some authors used an algorithm prior to bias adjustment. Cannon et al. (2015); Cannon (2018); Mehrotra et al. (2018); Vrac et al. (2016) corrected the number of wet days by





replacing the zero precipitation days with a trace amount (below 0.05 mm). Vrac et al. (2016) called this method "Singularity Stochastic Removal" (SSR) and they provide a more detailed explanation. This allows to use all days for bias adjustment, even the dry days, where otherwise zeros could cause problems during the bias adjustment when dividing by zero. After bias

adjustment values below the trace amount are reset back to zero.

For our data, SSR leads to a significant improvement when bias adjusting very dry model runs. Comparing the errors of SSR and the piecewise trapezoid method for precipitation sums and wet days, the latter one has either similar errors or slightly outperforms SSR.

### 3.4 Empirical Quantile mapping (QM)

We then compared the performance of EQA with other methods. One of them is the traditional QM in a non-parametric form, which is widely used (e.g. Piani et al., 2010; Themeßl et al., 2011; Teng et al., 2015; Gutiérrez et al., 2019; Maraun et al., 2019; Widmann et al., 2019). Quantile mapping in its original form is usually written as (Li et al., 2010; Themeßl et al., 2011):

$$x_{corr} = F_{o-c}^{-1}(F_{m-c}(x_{m-f})) \tag{11}$$

where F is the (in our case empirical) CDF of either the observations (o) or model (m) for a historic (calibration period) climate

(c) or future period (f). This QM can not produce values that are outside the observed range. In the context of climate change, new extremes are considered via a simple extrapolation: For values that are above or below the most extreme values found in the observations, a constant correction of the last value is applied (Boé et al., 2007). For example, if the highest temperature found in the historical model is 34 °C and the highest value in the observations is 36 °C, a correction value of +2 °C is applied to all future model values above 34 °C. QM including extrapolation is written as:

$$x_{corr} = F_{o-c}^{-1}(F_{m-c}(x_{m-f})) + \underbrace{x_{m-f} - F_{m-c}^{-1}(F_{m-c}(x_{m-f}))}_{\text{Extrapolation term}} \tag{12}$$

This formula is used in this work and is part of the code in the pyCAT module for Python. The extrapolation term is zero when $x_{m-f}$ lies within the range of historical model values. Comparing Eq. 12 with Eq. 1 shows that QM calculates the CDF from the model in the historical period ($F_{m-c}$), whereas EDCDFm (and therefore EQA) use data from the period where the correction is applied on ($F_{m-f}$).

### 330 3.5 Scaled distribution mapping (SDM)

The second bias adjustment method that EQA is compared with is the parametric SDM based on QDM (Switanek et al., 2017). SDM is available via the pyCAT module for python. SDM is a parametric method. For precipitation, gamma distribution can be selected. The parameters for the gamma distribution are found iteratively via the maximum likelihood function which can be computationally expensive. In our work, we observed the SDM script to be more than one order of magnitude slower than

EQA.





Tests showed that the fitting is sometimes defective and results in errors when the corrected model data is compared with the observations (see Fig. 2). Hence, the SDM script is not always able to reproduce the past climate by correcting the model according to the observations.

Therefore, we generated several versions of SDM. For this work, we improved the fitting of the gamma functions by adding

initial guesses to the fitting function. According to the methods of moments (Thom, 1958; Wiens et al., 2003), the initial guess for the scale parameter $\theta$ for the gamma distribution is defined as:

$$\theta = \frac{\text{Var}(X)}{\bar{X}} \tag{13}$$

where $X$ is the data to be fitted and $\bar{X}$ is the mean of the data. Optionally also the shape parameter $k$ can be used for the initial guess as

$$k = \frac{\bar{X}^2}{\text{Var}(X)}. \tag{14}$$

We used four different versions of SDM which are as follows:

**SDM(raw):** This is the version of SDM as presented in Switanek et al. (2017). SDM(raw) lacks the correction of wet days, if the model has too few wet days.

**SDM(0):** In addition to SDM(raw), corrected wet days are interpolated to the expected number of wet days which corrects a

wet day bias. This algorithm was provided by the authors of Switanek et al. (2017).

**SDM(1):** In addition to SDM(0), the shape parameter $k$ is used as an initial guess for the gamma distribution of precipitation.

**SDM(2):** In addition to SDM(0), both shape parameter $k$ and scale parameter $\theta$ are used in the initial guess.

## 4 Results

EQA, QM and SDM are evaluated in terms of three demands expressed at the end of Sect. 1.

### 4.1 Demand (1): Conservation of historical climate

The four versions of SDM are compared with non-parametric QM and EQA to show the biases that are introduced by the bias adjustment methods themselves. We already showed the biases in the ÖKS15 and STARC-Impact data (Fig. 2). To reproduce some of the biases, we used the smoothed observational data as produced in Sect. 2. Depending on the method of bias adjustment, even after correction the bias can be significant (Fig. 5). Figure 5a is the observed average annual precipitation (OBS),

where the impact of small scale spatial patterns like valleys, mountains and windward and leeward side can be seen. Fig. 5b is the artificial smoothed model, but otherwise very similar to OBS. The only difference is the spatial resolution between OBS

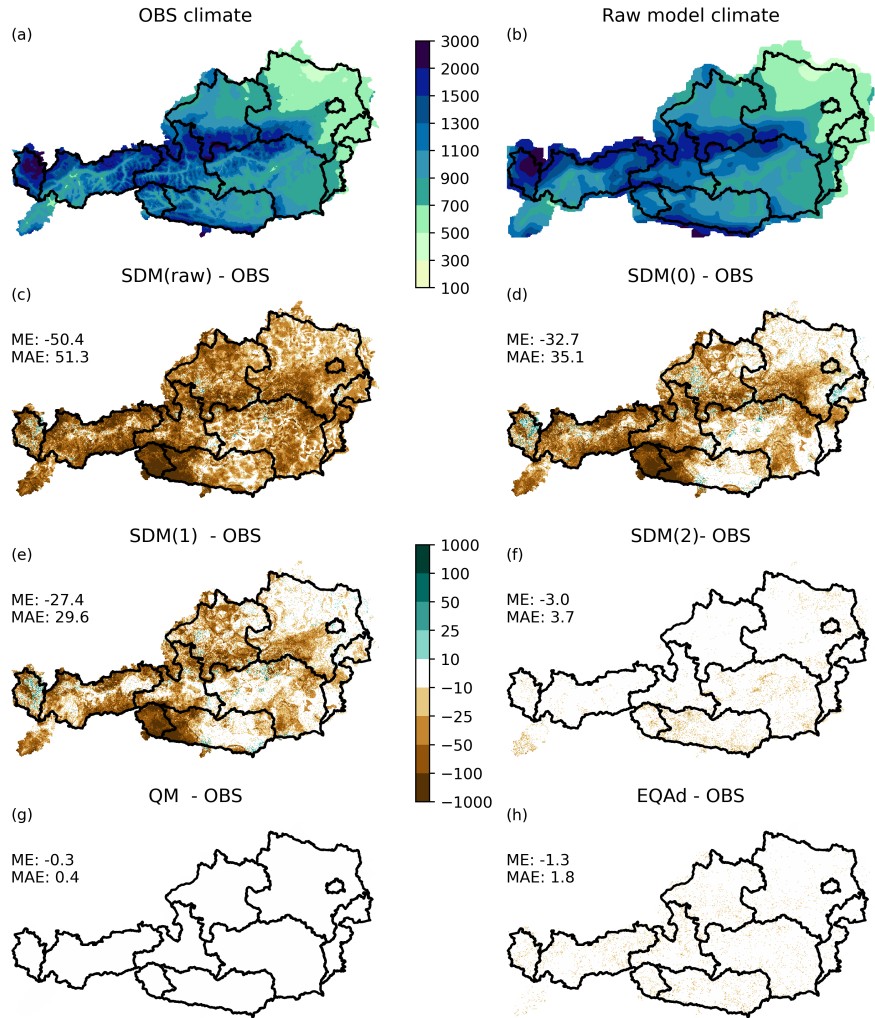

**Figure 5.** Bias adjustment of precipitation data. The model is produced by smoothing OBS. (a): Observational annual precipitation. (b): Raw model annual precipitation. (c)-(h): Difference in annual precipitation (model minus observational data) in mm for (c) SDM(raw), (d) SDM(0), (e) SDM(1), (f) SDM(2), (g) QM and (h) EQAd. ME: mean error. MAE: mean absolute error.

and the model. Figure 5c-h shows the difference of the mean annual precipitation of the bias adjusted model data minus the mean annual precipitation of the OBS.

Figure 5c uses SDM(raw) which produces the largest errors with a mean absolute error of 51.3 mm in annual precipitation.
The difference in annual precipitation exceeds 100 mm in parts of East Tyrol and Carinthia (southwestern parts of Austria). The errors produced by SDM(0) and SDM(1) (5d and e) are considerably smaller. The smallest errors are produced when using SDM(2), where the error is 10 mm or less in most parts of Austria with a mean absolute error of 3.7 mm per year.





For comparison, the model data was corrected with other methods such as QM where the error is close to zero. This is because empirical QM calculates exact empirical CDFs of both model and OBS which produces very accurate results in the reference period. After bias adjustment with EQAd there is small remaining error which we assume to be caused by the empirical CDFs which are defined at 100 discrete values. Similar errors have been found by Potter et al. (2020) and Charles et al. (2020). These small errors can be neglected, because the exact shape of the empirical CDF depends on the choice of the reference period. A shift of the reference period will also change the shape of the CDF, especially at the extreme ends. So, the almost non-existing error of QM is only a false accuracy as we can not expect that errors outside the reference period will be equally small when extremes change under a changing climate.

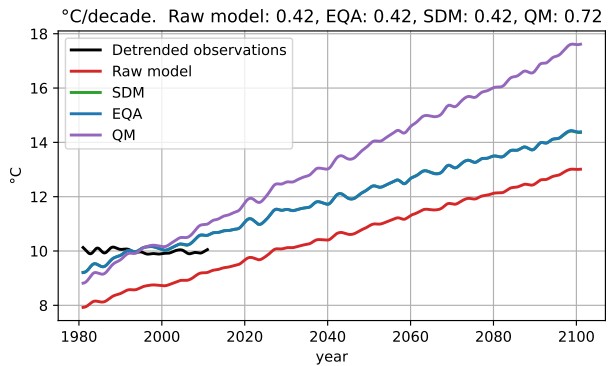

**Figure 6.** Running means of temperature data of detrended observations, the raw model and 3 different bias correcting methods (QM, SDM and EQA). SDM and EQA are almost identical. On top: average linear trends in °C per decade (1981-2100) for each bias adjustment method. The data is identical to the data used in Fig. 3.

### 4.2 Demand (2): Climate change signal (CCS)

Demand (2) for EQA is that the CCS of the raw climate model should not be altered. As stated by Maraun (2016), methods like standard QM can add a systematic error to the temperature CCS where the CCS is defined as an absolute value. Therefore we calculated the CCS for all three methods for temperature. As temperature data, we used the artificial data as generated in Sect. 3.1. The corresponding CDFs are shown in Fig 3. In Fig. 6 we show the smoothed mean annual temperature for the detrended observations, the raw climate model and for the bias corrected model data. The temperature of the raw model shows an increase of 0.42 C each decade. The bias adjustment methods SDM and EQA reproduce the exact same trend, and are thus able to exactly conserve the CCS. In contrast, QM inflates the climate change signal with a linear trend of 0.71 C per decade. We also tested all three methods with non-linear trends, where QM tends to inflate or deflate the CCS (not shown) while SDM and EQA keep the CCS unchanged.

For precipitation, the CCS is defined as a relative value as shown in Eq. 7 and 8. The relative CCS is greater than 1 in case of more precipitation in the future. In Sect. 2 an artificial dry model was produced by drying OBS. Figure 7a shows



the mean annual precipitation of the model for the historical climate (much drier than observations), and Fig. 7b shows the mean annual precipitation of the model for a future period which is even drier. Figure 7c-f shows the CCS error of the bias

adjustment methods according to Eq. 9. As before, SDM generally underestimates the CCS (Fig. 7c) while EQAd (without the CCS correction) and QM generally overestimate the CCS (Fig. 7d and 7e). However, the mean absolute error of EQAd (1.9 %) is smaller than those of SDM (4.3 %) and QM (3.3 %). In Fig. 7f with EQAd the CCS of the annual precipitation is forced to match the raw model CCS via Eq. 10, therefore the error is almost 0 %.

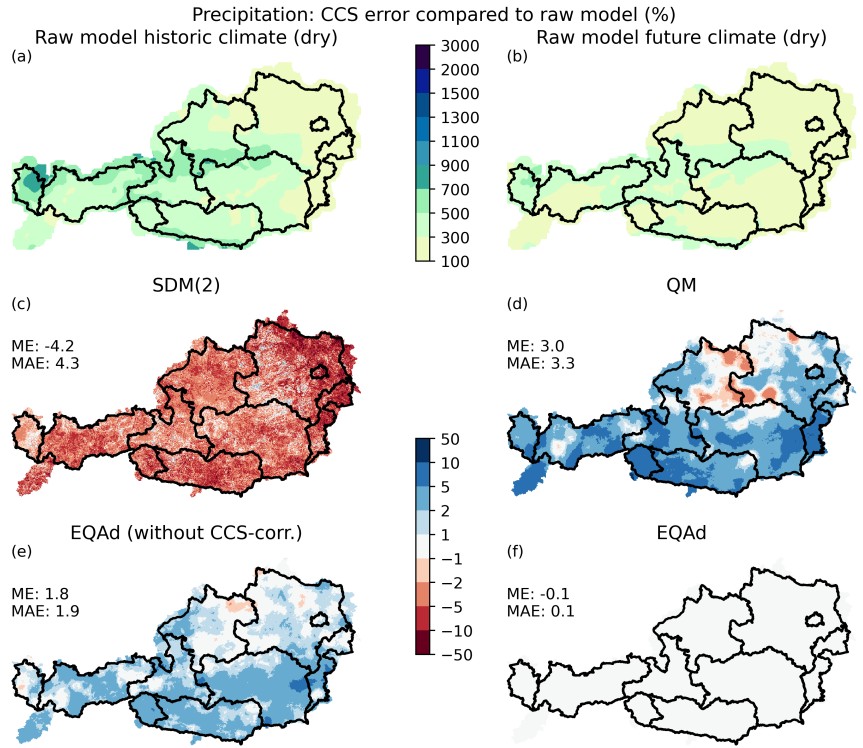

**Figure 7.** Error of CCS compared to CCS of raw model (Eq. 9). (a): Raw model annual precipitation in historic period (mm). (b): Raw model annual precipitation in future period (mm). (c) SDM, (d) QM, (e) EQAd without additional correction of the CCS and (f) EQAd. A perfect bias adjustment equals 0 %.

## 4.3   Demand (3): Wet day frequency in dry models

As already discussed in relation to Fig. 5, parametric methods do not always reproduce the observational climate. Furthermore, very few bias adjustment methods accurately bias correct climate models with a distinct dry bias. We compare SDM, QM and EQAd using the artificial dry model data (Fig. 8b). The difference of the model data corrected with SDM(2) minus OBS shows quite good results, but overall the corrected data shows a slight wet bias that exceeds 25 mm in some grid cells (Fig. 8c). The area mean annual bias of SDM(2) is 6 mm. QM corrects the precipitation for already existing wet days but cannot add wet





days. Thus, there is still a dry bias after bias adjustment (Fig. 8d), where the area mean is -48.5 mm. EQAd shows a similar
pattern (Fig. 8e) as it is almost identical to QM in the historical period, with the only difference that EQAd uses 100 discrete
percentiles and QM uses all values for the CDFs. EQAd adds wet days and is able to accurately reproduce climatological
precipitation sums in the historical period with an average annual precipitation bias of only 2.5 mm.

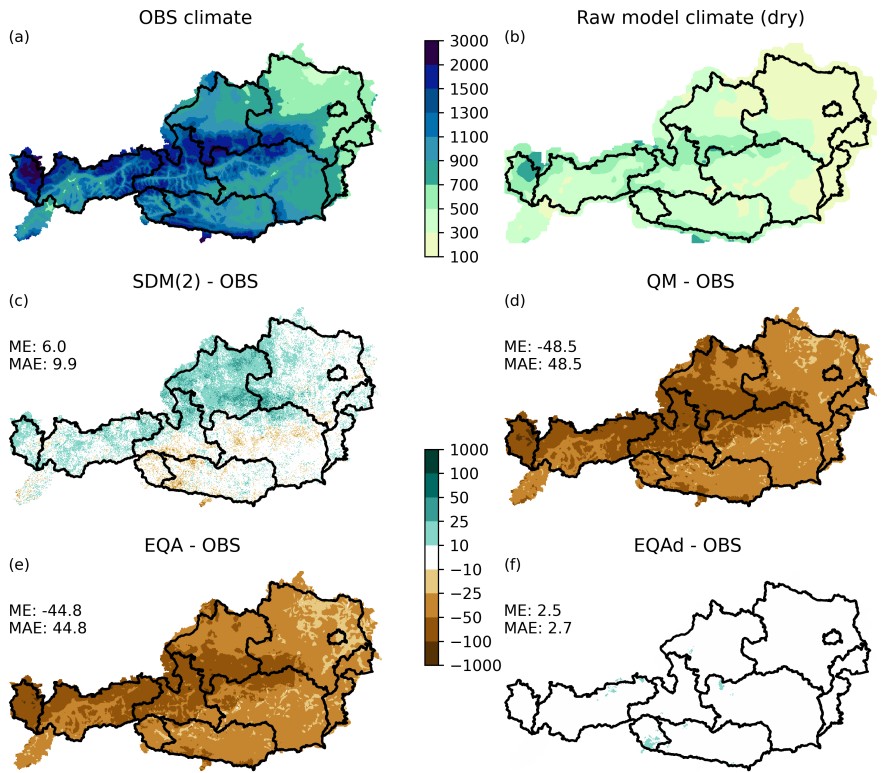

**Figure 8.** Climatological annual precipitation (mm) sum in the historical period for dry model data. (a): OBS annual precipitation. (b): Raw
model (dry) annual precipitation. (c)-(f): Difference in annual precipitation (model minus observed data) in mm for (c) SDM, (d) QM, (e)
EQA and (f) EQAd.

Figure 9 shows the number of precipitation days. The average number of precipitation days per year is much higher in the
observations (Fig. 9a) than in the model (Fig. 9b). The difference of Fig. 9a and Fig 9b is the error of precipitation days per
year of the raw model.

The parametric SDM(2) produces too many new precipitation days (Figure 9c). The average annual wet day bias is +15.5
days. Both the non-parametric QM and EQA (Figure 9d and e) cannot change the number of wet days without further modifi-
cations, so the average annual wet day bias of -69.9 days of the raw model is unchanged. EQAd (Figure 9f) performs best of
all methods with an average bias of only -2.3 wet days per year. Only very few grid cells exceed a wet day bias of +10 or -10
days.

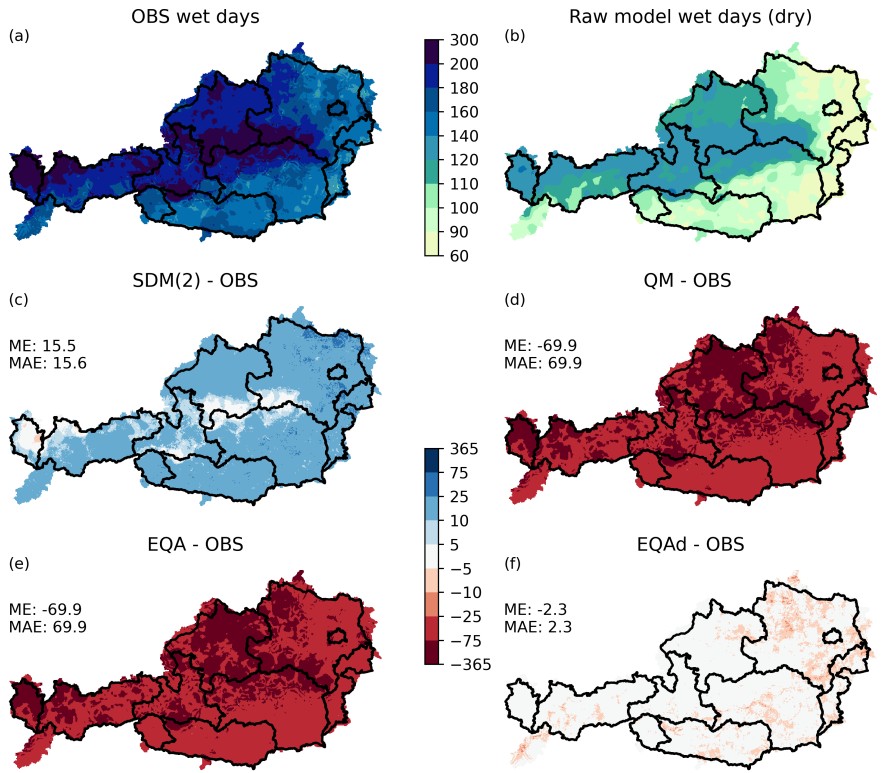

**Figure 9.** Wet days per year ($\geq 0.1$ mm) in the historical period for dry model data. (a): Annual wet days in OBS. (b): Raw model annual wet days. (c)-(f): Difference in annual wet days (model minus observed data) for (c) SDM, (d) QM, (e) EQA and (f) EQAd.

## 5 Conclusions

Statistical bias adjustment methods are widely used to improve direct model output from climate models but cannot fully remove all model errors. The adjusted data is often used as input for climate impact studies where biases can significantly alter the impact analysis, so one has to be aware of the limitations of the bias adjustment methods. We compared three methods (Empirical QM, SDM and EQA) which all adjust the statistical distribution of meteorological variables. We evaluate the methods with the three demands formulated in the introduction for synthetic climate data to show that errors can originate from the bias adjustment method and not only from climate models.

Table 2 summarizes our main results on two of the three demands. The tested bias adjustment methods are grouped by how the CDFs are calculated (empirical or parametric) and by whether the bias is assumed to stay constant at quantiles or at a specific value of a variable. We assume that SDM can be seen as a representative of parametric methods in general because the errors introduced with SDM are mainly due to the fitting of functions.

– Demand (1): EQA and QM are capable of statistically correcting the model's past climate to fit the observations accurately. This is mostly due to the fact that they are non-parametric methods, i.e. they use empirical distribution functions





instead of fitted functions (for variables like temperature, precipitation etc.) which allows the CDF to follow any possible
       shape (Table 2). The fitting of functions (SDM) will always produce errors which can be minimized with a good fitting
       algorithm (Fig. 5). Also parametric approaches require knowledge about the statistical distribution of a meteorological
       variable in order to choose a suitable distribution function.

       – Demand (2): EQA and SDM barely enhance or suppress the mean CCS in contrast to traditional QM, i.e. they explicitly
reproduce the same CCS as in the raw model. For additive EQA and SDM (e.g. for temperature), this is valid without
         any limitation (Fig. 6). For multiplicative EQA and SDM (e.g. for precipitation), the CCS is defined as a ratio between
         historical and future climatological mean. EQA preserves this ratio at every quantile (Table 2). However, in general the
         relative CCSs of monthly and annual means differ from the ratios at quantiles. Depending on the application, a decision
         has to be made either to conserve the relative CCS at quantiles or at means. In the latter case, an algorithm after bias
adjustment corrects means to match the raw model's CCS.

       – Demand (3): The third demand is not mentioned in Table 2, because neither SDM(raw), QM or EQA by themselves are
         able to correct models with too few wet days, if applied multiplicatively. SDM(0), SDM(1) and SDM(2) interpolate the
         wet days to the expected number of wet days which should correct the bias. Fig. 9c shows that there is still a wet day
         bias after correction with SDM(2), though with a positive sign (too many wet days). The reason for this positive wet bias
is still in discussion. We suspect that it might be caused by the fitting of gamma functions to the CDFs which introduces
         new errors. As an alternative, we provided an algorithm that follows the bias adjustment and adds additional wet days
         in order to reproduce the observation's precipitation sums and wet day frequency (Fig. 8f and 9f). To indicate the use
         of this algorithm we added the letter d (EQAd). As a supplementary method the algorithm can be applied after any bias
         adjustment method and is therefore independent of EQA. Other methods, like SSR (Vrac et al., 2016) are also capable
of correcting a dry bias and can be used as substitute of EQAd. In the case of a model having too many wet days, the
         wet day frequency is automatically corrected with all three methods.

**Table 2.** As in Table 1. (1) and (2) refer to the demands formulated in the introduction.

|  | Parametric | Non-parametric / Empirical |
|---|---|---|
| Bias at fixed quantile / Trend preserving | **SDM:** (1) no (2) yes (additively), only at quantiles (multiplicatively) | **EQA:** (1) yes (2) yes (additively), only at quantiles (multiplicatively) |
| Bias at fixed value / Trend altering | **(not tested)** (1) no (2) no | **Empirical QM:** (1) yes (2) no |



A good performance of the corrected data in any of the three demands is crucial, as it is used as input for further impact studies. Impact models (e.g. plant growth models) are often calibrated with bias corrected historical meteorological data from a climate model. The focus of impact studies often lies on the CCS. If an impact model is calibrated with inaccurate meteoro-
logical data in the historical period, the impact of climate change can lead to wrong conclusions even if the CCS is accurate.

To sum up, EQAd is able to reproduce the observation's statistical distribution, is able to preserve the raw model's CCS and can add wet days if necessary because of a supplementary algorithm.

EQAd along with many other methods corrects each grid cell independently. We showed that the spatial patterns of the corrected data matches the observations for long term means. However, spatial patterns of smaller time scales (e.g. a season,
a month or a single day) are only corrected to a limited extent. For a more accurate representation of temporal or spatial correlations, other methods have to be applied (e.g. Nguyen et al., 2016, 2017; Mehrotra and Sharma, 2016; Mehrotra et al., 2018; Mehrotra and Sharma, 2019; Volosciuk et al., 2017; Cannon, 2018).

*Competing interests.* The authors declare that they have no conflict of interest

*Acknowledgements.* This work was partially supported by FORSITE (Waldtypisierung Steiermark - Erarbeitung der ökologischen Grund-
lagen für eine dynamische Waldtypisierung), funded by the government of Styria in Austria. The precipitation data set SPARTACUS was generously provided by ZAMG. We also thank Copernicus Land Monitoring Service as part of the European Environment Agency (EEA) for the topography data. Finally, we thank Douglas Maraun for his helpful comments.



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
