# Peer review of "Evaluating quantile-based bias adjustment methods for climate change scenarios"

_Hydrology and Earth System Sciences, 2021_

## Referee Comment (RC2)

Review of

'Evaluating quantile-based bias adjustment methods for climate change scenarios'

by F. Lehner, I. Nadeem and H. Formayer

**Recommendation: minor revisions**

This manuscript presents the 'Empirical Quantile Adjustment' (EQM) method for statistical bias correction of climate model output, including a version that allows to increase the number of dry days (EQAd). The EQA method is a development of existing versions for bias correction, while the dry-day correction is a new approach. The manuscript compares EQAd with two frequently used alternative approaches to bias correction, with respect to the representation of the mean, of the climate change signal (CCS), and of the number of dry days. These criteria reflect practical requirements well.

It is found that EQAd performs as well or better than the other methods on each of the criteria. This is not surprising because EQAd has been designed to do well with respect to all of these criteria. The value of the manuscript is that it demonstrates that other methods have some practically relevant shortcomings, and that it is possible to construct a bias correction method that addresses these.

I have reviewed a previous submission by the authors on this topic, which was not published, mainly because the bias correction approach taken was not justified well enough. The new manuscript is substantially improved, much more systematic, and now explains most conceptual and technical aspects very well. In addition, it provides a systematic overview on the various approaches to bias correction with a good discussion of their structural differences.

This paper is now a very useful contribution to the development and understanding of bias correction methods. It is concisely written and it was a pleasure reading it.

The points listed below should be further clarified. I recommend publication after they have been addressed.

**Specific comments**

1)
The authors frequently mention that EQA and similar methods are based on quantiles, that the biases at quantiles do not change over time etc., whereas QM is said to be based on specific values. This terminology is not wrong, but I don't think it is the best choice to reduce confusion. Many researchers would presumably consider QM to also be based on quantiles (hence the name). The key question is from which distribution the quantiles are determined for a given value in the future. As mentioned by the authors in the last sentence of section 3.4 QM determines the quantile for a given future value from the calibration distribution.

This point should be clarified already in the introduction when the different approaches are discussed.

2)
The point above also applies to the discussion of stationarity in lines 63-68. QM assumes stationarity of the bias for each quantile with quantiles derived from the calibration distribution, regardless of whether a value is taken from the calibration or the future periods. EQA and similar methods assume a stationary bias for each quantile, but with the quantiles for a value from the future period derived from the future distribution. As discussed in the manuscript, the latter is for additive corrections equivalent to assuming a stationary bias in the mean, but this is not the case for multiplicative methods.

3)
Fig 3. is very helpful, but I am not convinced it is in the right place in the paper. The figure illustrates the two basic options for distribution-based bias correction. This is a general issue and not specific to EQA. The related discussion could be part of the method review in the introduction, or a new subsection could be included at the beginning of the method section that systematically discusses the two options (this section could also address the next two points).

The very large changes in the CCS introduced by QM in Fig. 3 are essentially caused by the fact that the raw CCS is large compared to the standard deviation of the distributions, and that the standard deviations of the model and the observations are quite different. This is useful to clarify the argument, but it might be interesting to also consider a less extreme example.

4)
In section 3.1 the authors attempt to explain why the EQA and other approaches that determine the quantile for a future value from the future distribution are better justified than the QM approach. The argument is based on the claim that RCMs are 'able to predict a ranked' category, and on 'a specific weather pattern will have different absolute values in the future but the same quantile'. The arguments are not clear to me. Ranking is an ordering of a set of values. Which values are considered here? What means a weather pattern has the same quantile?

These arguments should be clarified and may add interesting aspects, but views on these issues might differ. I don't think these are the key arguments in support of bias correction methods that use future distributions.

5)
In my understanding the most important justification for using EQA and similar methods is that it preserves the CCS for additive corrections, and with the modification introduced in the paper also for multiplicative corrections.

As I said in my previous review it is far from clear that the CCS should be preserved. Nevertheless, preserving the CCS might in many applications be a more sensible approach than altering it in a rather uncontrolled way by using QM. It is good that the paper makes it clear now that preserving the CCS is a choice, not an a priori given desired property. I think it would be good to emphasise that if researchers decided to retain the CCS the EQA method provides a bias correction method that does this and has additional useful properties.

6) Line 213:, $F\_100$ are not 'the 100 CVs to correct the model data', they are the percentiles for which the CVs are defined through eqn. 3.

7) Lines 221-222: What is called 'parameters' here should be 'variables'. A variable is a number that changes, for instance in time and space, and specifies the state of a system. A parameter is a number that can take different values and once specified specifies the system itself (for instance parameterisations in dynamical models or parameters in statistical models).

I know that meteorologist tend to use the former when it should be the latter, but throughout most of the paper variables are correctly called variables, and this should be done everywhere.

8) Eqns. 5 and 6: The variable names for the ranked variables should be different from those for the unranked variables. This could be done for instance by adding a ',r' to the subscripts.

9) Lines 246-248: The explanation for step 6 is unclear.

10) Line 300: 'in future' should be 'in the future'

11) Fig. 5. It would be informative to add a panel that shows the difference between the raw model and the observations.

12) Line 378: I don't think one should use the formulation 'error in CCS', because the true CCS is unknown. It is better to say that QM modifies the CCS from the raw model, and that the specific modification might be difficult to justify.

13) Figs. 5 – 9. The interval boundaries for the colour bars are neither linear nor logarithmic, and some of the colours are difficult to distinguish. Is there a clear reason for the choice of the intervals? Can they be defined more systematically?

14)
There are several sentences where 'which' is used in situations where it should be 'that'. The former should be used if additional information is added, the latter if a defining property is stated.

---

## Author Comment (AC1)

*This article seeks to present a new method for bias correction of climate model outputs. Also, it seeks to present an intercomparison of bias correction between three available methods, one of which is the new method. Bias correction is a very important aspect of future planning for climate change given the strong limitations of raw model outputs for practical hydrological decision making. Thus, articles suggesting method improvements, or with insightful intercomparisons, should be encouraged.*

*However, in this case, I fear a lot of work is required to raise the material up to publication standard within HESS. I have a number of concerns, none of which are specifically about the technical details of the method, but some of which build on technical concerns raised by the reviewers of the original pre-print (HESS-2020-515). The concerns are listed below.*

As referee #2 was also the more critical referee during the first submission, we want to emphasize here that his/her concerns have been addressed and solved.

*1. Detrimental dual focus (aka what is the paper actually about?)*

*In response to the reviewer comments, the authors have elected to increase the prominence of the method intercomparison and decrease the prominence of the new method. For example, the title is now 100% about the evaluation and doesn't mention the new method, and the abstract doesn't go into details about what is new in the method, rather, it just says "we present a method". In contrast, the body text follows the original manscript and is still focussed on the new method. So the paper has a dual focus. Unfortunately, it doesn't really work to have a paper about two things—each detracts from the other, leaving both unsatisfactory. So the authors need to decide: if the paper is about the new method, then the evaluation is subsidiary and should be specifically designed to justify why the new method is an improvement. Alternatively, if the paper is about the intercomparison/evaluation, the new method should \*not\* be mentioned at all.*

We are aware that the focus of the manuscript was not clear enough in the first submission and have worked a lot on this issue. The goal of this paper is to find a bias correction method that fulfils three well-founded demands that a highly relevant for climate impact scientists. Conversely, we also discuss the shortcomings of two other methods, which are representatives for larger groups of methods.

Also, in the existing literature it is not pointed out clearly that our demand (2) depends on how the quantile correction is applied. If the quantiles are calculated only with the historical values like in traditional QM, which means the bias adjustment is fixed to constant values of the variables, the raw CCS is changed. This we already mention, but it will be made clearer in the introduction, in the text for Fig. 3 and in a new subsection 3.1. as mentioned in our response for 3) of referee #2. We consider this aspect as valuable contribution to the community.

To meet your concerns and in agreement with the comments from the other referee, we move some parts of 3.1 to a new subsection (See 3.) from referee #2 where we discuss the conceptional differences between QM and EQA and all methods, that are in the same groups according to Table 1. So in the end, the description of EQA is strongly shortened and only serves to document our method for the readers so that it can be reproduced. This should also reduce the impression that we intend to introduce a completely new method.

*2. Is the method an advance?*

*If the authors elect to focus on the method, the next question is whether the method is an advance. Prompted by reviewer 1, the authors admit that "Equation 2 in Li et al. (2010) is the correct mathematical*

*description for our method when correcting temperature" and that "for precipitation, our method is almost equivalent to PresRAT". However, "the important difference to these methods is that our method is stricly empiric/nonparametric". Given this description and perusing the HESS manuscript types at https://www.hy-drology-and-earth-system-sciences.net/about/manuscript_types.html, I suggest that this kind of change is more appropriate for a "Technical Note" and does not constitute a theoretical or practical advance worthy of a research article. If this path was taken, the material would need to be significantly reduced to fit with the requirement of "a few pages".*

The bias adjustment method itself is an empiric application of existing methods to meet the three specified demands. This may not be sufficient for being called a new method, thus we avoided to call it "new". However, we decided to use a new name for it, as it is not completely identical to any other method:

For additive bias adjustment for e.g. temperature, EQA is equivalent to EDCDFm and QDM (Cannon et al., 2015), with the emphasis that we calculate purely empirical CDFs. For precipitation (multiplicative bias adjustment), EQA is equivalent to PresRAT with the difference that EQA is empirical.

A novelty is the combination of the above said, the fundamental discussion about them, the focus on complex terrain with strongly varying meteorological fields and that we compare the methods with synthetical model data to show how the methods deal with extremely dry models.

*3. Are the methods being compared the right methods to compare?*

*The answer to this question depends on the choices made at point 1. If the author focus is to introduce a method that builds upon existing methods (as per point 2 above), then the most obvious point of comparison is the parent methods. Thus, the comparison should include EDCDFm, PresRAT, and any other methods the new method inherits from. This is the only way to determine whether the new method helps or hinders, and in which context this occurs. On the other hand, if the authors elect to focus on method evaluation/intercomparison, then I think some changes will be required to make the evaluation more novel. This will require reviewing what other literature tries to do this (with which I confess I am not fully familiar) and then ensuring there is something different (and substantial) about this intercomparison - perhaps it's the number of methods being compared, or that their types have never been compared, or perhaps it's the study area that's novel (Austria). In any case, it seems likely that more than two methods will need to be compared, which translates to more work for the authors I'm afraid.*

The choice of methods depends on what is intended to show. We wanted to show the shortcomings of widely used methods. Thus, our selection had two reasons:

- We arranged bias adjustment methods in four groups in Table 1. We chose one method out of three of these groups. The forth group (trend altering and parametric) was not relevant for this comparison, as it unites negative features of two other groups, so we skipped this group.
- The chosen representatives of the groups should be relevant for the impact research community of the alpine area, which means that is should have been implemented (several times) in Austria or Europe. This applies to QM and SDM. SDM was used for a couple of projects in Austria, where the data is publicly available for climate impact studies.

EQA, QM and SDM all represent their groups in Table 1.

*4. What literature review is required and how should the study motivation be framed?*

*In the case where a slight tweak is made to existing methods, there is no need for a lengthy literature review espousing the benefits of the parent methods. We can assume that the case was argued back when the*

*original methods were published, and thus all that is required is a short summary of the benefits (or, if debated, of either side of the debate). This is why a "Technical Note" can be (indeed, must be) so short. On the other hand, if the focus is on intercomparison, then the authors must provide a general review of all bias correction methods, a review of other intercomparison studies, and a justification for what this study is adding to those existing studies. Unfortunately, the current introduction does none of these things well. It is quite long and lacks narrative and structure.*

There is a structure in the introduction, which was also suggested by the second reviewer from the first submission. Every paragraph has a specific topic/focus. As it is not possible to make headlines in the introduction, these are not directly visible. The introduction should serve the following purposes:

1. It introduces our three demands and justifies them.

2. It introduces existing methods that may be relevant for meeting the three demands.

3. It should give the reader all the necessary background information to understand Table 1. (parametric/empirical, CCS, stationarity of biases)

4. It briefly discusses comparison studies.

However, we agree that the introduction is quite long and there is still room for the improvement of the structure. If we are offered the chance to stay in the reviewing process, the introduction will be rearranged to serve the purposes in the right order. We will shorten the introduction, e.g. by moving lines 96-107 to the end, where it could be added to a new "discussion" section. These lines were strongly influenced by referee #1 of the first submission, who wanted more bias adjustment papers with hydrological applications in mind. Also, we will move the three demands more towards the beginning of introduction, as they are the base for the whole following literature review. Additionally, single sentences can also be removed here and there without omitting too much information.

*I could go into greater detail on some further technical aspects of the paper but I feel there is not much point until these big, overarching questions are settled, since they will impact many aspects of the manuscript. I would be happy to review future versions and give more detailed comments then.*

*In general, this manuscript has the feel of funded applied research being turned into a paper as an afterthought. Funded applied research is very worthwhile in and of itself but to add something to the academic literature there must be something substantive that is new, novel or insightful, and not every practical project has these aspects.*

You are right in this aspect that we are partly paid by funded applied research projects. We have already implemented our method in projects. However, a literature screening did not explain satisfactorily the skills and limitations of the methods and which are suitable for our needs. This is why we decided this is worth a publication.

*I feel that I should end on some positives. Notwithstanding my criticisms of structure, the article is quite well written, with a good standard of English, well presented (particularly the figures) and with good attention to detail. I wish the authors well in to alter/add/refine the material so that it does eventually appear in the published literature.*

Thank you for pointing out the positive aspects in your opinion.

---

## Author Comment (AC2)

As this referee has already reviewed the first submission of the manuscript, we are glad to read that the referee appreciates our changes to the script.

***Recommendation: minor revisions***

*This manuscript presents the 'Empirical Quantile Adjustment' (EQM) method for statistical bias correction of climate model output, including a version that allows to increase the number of dry days (EQAd). The EQA method is a development of existing versions for bias correction, while the dry-day correction is a new approach. The manuscript compares EQAd with two frequently used alternative approaches to bias correction, with respect to the representation of the mean, of the climate change signal (CCS), and of the number of dry days. These criteria reflect practical requirements well.*

*It is found that EQAd performs as well or better than the other methods on each of the criteria. This is not surprising because EQAd has been designed to do well with respect to all of these criteria. The value of the manuscript is that it demonstrates that other methods have some practically relevant shortcomings, and that it is possible to construct a bias correction method that addresses these.*

*I have reviewed a previous submission by the authors on this topic, which was not published, mainly because the bias correction approach taken was not justified well enough. The new manuscript is substantially improved, much more systematic, and now explains most conceptual and technical aspects very well. In addition, it provides a systematic overview on the various approaches to bias correction with a good discussion of their structural differences.*

*This paper is now a very useful contribution to the development and understanding of bias correction methods. It is concisely written and it was a pleasure reading it. The points listed below should be further clarified. I recommend publication after they have been addressed.*

***Specific comments***

*1)*

*The authors frequently mention that EQA and similar methods are based on quantiles, that the biases at quantiles do not change over time etc., whereas QM is said to be based on specific values. This terminology is not wrong, but I don't think it is the best choice to reduce confusion. Many researchers would presumably consider QM to also be based on quantiles (hence the name). The key question is from which distribution the quantiles are determined for a given value in the future. As mentioned by the authors in the last sentence of section 3.4 QM determines the quantile for a given future value from the calibration distribution. This point should be clarified already in the introduction when the different approaches are discussed.*

Thanks for this clarification. We will mention this after the paragraph in the introduction (line 62), where we discuss the methods that do not alter the CCS. Both the CCS discussion and whether the quantiles (CDF) are calculated from the calibration or from the future period are highly related to each other.

Also, we will change this in the lines 181-183 and in the conclusion around line 420.

*2)*

*The point above also applies to the discussion of stationarity in lines 63-68. QM assumes stationarity of the bias for each quantile with quantiles derived from the calibration distribution, regardless of whether a value is taken from the calibration or the future periods. EQA and similar methods assume a stationary bias for*

*each quantile, but with the quantiles for a value from the future period derived from the future distribution. As discussed in the manuscript, the latter is for additive corrections equivalent to assuming a stationary bias in the mean, but this is not the case for multiplicative methods.*

This is an interesting perspective, that will be included in the above-mentioned paragraph.

*3)*

*Fig 3. is very helpful, but I am not convinced it is in the right place in the paper. The figure illustrates the two basic options for distribution-based bias correction. This is a general issue and not specific to EQA. The related discussion could be part of the method review in the introduction, or a new subsection could be included at the beginning of the method section that systematically discusses the two options (this section could also address the next two points). The very large changes in the CCS introduced by QM in Fig. 3 are essentially caused by the fact that the raw CCS is large compared to the standard deviation of the distributions, and that the standard deviations of the model and the observations are quite different. This is useful to clarify the argument, but it might be interesting to also consider a less extreme example.*

As the introduction is already very long, a new subsection "3.1 Definition of quantile-based methods" in section 3 makes more sense. In this subsection, we will discuss Fig. 3. We also can include new subplots in the figure where the CCS is smaller than the standard deviation of the data.

*4)*

*In section 3.1 the authors attempt to explain why the EQA and other approaches that determine the quantile for a future value from the future distribution are better justified than the QM approach. The argument is based on the claim that RCMs are 'able to predict a ranked' category, and on 'a specific weather pattern will have different absolute values in the future but the same quantile'. The arguments are not clear to me. Ranking is an ordering of a set of values. Which values are considered here? What means a weather pattern has the same quantile? These arguments should be clarified and may add interesting aspects, but views on these issues might differ. I don't think these are the key arguments in support of bias correction methods that use future distributions.*

We postulate that the bias of a climate model is correlated to the modeled weather pattern. If we assume that the frequency of weather patterns does not change significantly over time, this means, that certain quantiles of temperature are linked to certain weather situations. Because QM corrects absolute values, regardless of the underlying weather conditions, trends are modified. Thus, if one wants to preserve trends, one must at least implicitly account for weather conditions. Now, we argue that EQA and similar methods do this implicitly, assuming that weather situation frequencies change little and that biases are primarily weather situation dependent, not absolute value dependent: a weather situation in the future would then have a higher temperature value accordingly, but still the same quantile. An example:

A cold winter day in Austria is related to moderate northeasterly flows and usually high atmospheric pressure with low wind and clear sky conditions. Cold translates to a low quantile for temperature. In future, the error of the model in this weather situation is assumed to stay constant. This weather situation will still translate to a low quantile in the future distribution, however the absolute temperature values are higher (respectively the corresponding quantile calculated from the historical distribution is higher).

If the referee agrees, we will adapt the lines 177-186 to the above said. Also, this is background information and will be shifted to the new subsection 3.1 and the old subsection will be limited to the description of the application of the EQA.

*5)*

*In my understanding the most important justification for using EQA and similar methods is that it preserves the CCS for additive corrections, and with the modification introduced in the paper also for multiplicative corrections.*

*As I said in my previous review it is far from clear that the CCS should be preserved. Nevertheless, preserving the CCS might in many applications be a more sensible approach than altering it in a rather uncontrolled way by using QM. It is good that the paper makes it clear now that preserving the CCS is a choice, not an a priori given desired property. I think it would be good to emphasise that if researchers decided to retain the CCS the EQA method provides a bias correction method that does this and has additional useful properties.*

We agree. That is why in the introduction we will point out more clearly, that preserving the CCS is a choice of the researcher and mention this also in the new subsection 3.1.

*6) Line 213:, $F\_100$ are not 'the 100 CVs to correct the model data', they are the percentiles for which the CVs are defined through eqn. 3.*

Will be changed to your suggestion.

*7) Lines 221-222: What is called 'parameters' here should be 'variables'. A variable is a number that changes, for instance in time and space, and specifies the state of a system. A parameter is a number that can take different values and once specified specifies the system itself (for instance parameterisations in dynamical models or parameters in statistical models).*

*I know that meteorologist tend to use the former when it should be the latter, but throughout most of the paper variables are correctly called variables, and this should be done everywhere.*

Thanks you for this clarification. It will be changed to "variables".

*8) Eqns. 5 and 6: The variable names for the ranked variables should be different from those for the unranked variables. This could be done for instance by adding a ',r' to the subscripts.*

This will be added to the variables.

*9) Lines 246-248: The explanation for step 6 is unclear.*

30 years of data is used for bias adjustment. Usually, climate models cover more than 100 years. Theoretically, it is possible to bias adjust 30-year periods that do not overlap, e.g. 1981-2010, 2011-2040, 2041-2070 etc. However, this will introduce inhomogeneities between the periods. This is why the 30 years of data is shifted in 10-year steps and only the middle 10-year period is actually kept as bias adjusted data.

This will be clarified in step 6.

*10) Line 300: 'in future' should be 'in the future'*

Done.

*11) Fig. 5. It would be informative to add a panel that shows the difference between the raw model and the observations.*

This is a good suggestion. The figure will be reshaped to a 3x3 plot which contains also the difference between raw model and observations.

*12) Line 378: I don't think one should use the formulation 'error in CCS', because the true CCS is unknown. It is better to say that QM modifies the CCS from the raw model, and that the specific modification might be difficult to justify.*

This will be changed. Also, in Fig. 7 we will change the "error" to "modification of CCS".

*13) Figs. 5 – 9. The interval boundaries for the colour bars are neither linear nor logarithmic, and some of the colours are difficult to distinguish. Is there a clear reason for the choice of the intervals? Can they be defined more systematically?*

The colors were chosen because they should be intuitive, e.g. yellow-green-blue for precipitation. However, we will make the bars less arbitrary. The new color bars will be more linear respectively logarithmic:

Annual precipitation sum: 100, 300, 500, 700, 900, 1100, 1300, 1500, 2000, >2000

Annual precipitation difference (Fig.5 and Fig. 8): <-80, -40, -20, -10, 10, 20, 40, 80, >80

CCS modification (Fig.7): <-8, -4, -2, -1, 1, 2, 4, 8, >8

Annual wet days (Fig.9): <90, 90, 100, 110, 120, 140, 160, 180, 200, >200

Annual wet days difference (Fig.9): <-80, -40, -20, -10, 10, 20, 40, 80, >80

*14)*

*There are several sentences where 'which' is used in situations where it should be 'that'. The former should be used if additional information is added, the latter if a defining property is stated.*

In our opinion, this refers to the following lines and will be changed: 4, 59, 109, 212, 288, 316, 346 and 416.